# Overweight and obesity among women at reproductive age in Cambodia: Data analysis of Cambodia Demographic and Health Survey 2014

**Samnang Um** [1]*, **An Yom** [1], **Jonathan A. Muir** [2], **Heng Sopheab** [1]

**1** School of Public Health at the National Institute of Public Health, Phnom Penh, Cambodia, **2** The Global Health Institute, Emory University, Atlanta, Georgia, United States of America

* umsamnang56@gmail.com

**Data Availability Statement:** Our study used the 2014 Cambodia Demographic and Health Survey (CDHS) datasets. The DHS data are publicly

## Abstract

Overweight and obesity increase the risk of cardiovascular disease, type 2 diabetes mellitus, hypertension, stroke, and some type of cancers, and maternal health globally. In Cambodia, the prevalence of overweight and obesity among women aged 15–49 years increased from 6% in 2000 to 18% in 2014, becoming a public health burden. We examined socio-demographic and behavioral factors associated with overweight and/or obesity among women of reproductive age (WRA) in Cambodia. We analyzed data from the 2014 Cambodia Demographic and Health Survey (CDHS) that used a two-stage stratified cluster sampling design. Data analysis was restricted to non-pregnant women, resulting in an analytic sample of 10,818 women. Multivariable logistic regressions were performed using STATA V16 to examine factors associated with overweight and obesity. Prevalence of overweight and obesity among non-pregnant women of reproductive age were 15.2% and 2.8% respectively. Factors independently associated with increased odds of overweight and/or obesity including women aged 20–29 years with adjusted odds ratio [AOR = 2.4; 95% CI: 1.6–3.6], 30–39 years [AOR = 4.6; 95% CI: 3.0–6.9], and 40–49 years [AOR = 6.6; 95% CI: 4.3–10.1], married women [AOR = 1.8; 95% CI: 1.3–2.7], urban residence [AOR = 1.3; 95% CI: 1.1–1.5], and women having at least 4 children [AOR = 1.7; 95% CI: 1.2–2.5]. The factors were associated with decreased odds of overweight and obesity: completed at least secondary education [AOR = 0.7; 95% CI: 0.6–0.8], agricultural work [AOR = 0.7; 95% CI: 0.5–0.8], and manual labor work [AOR = 0.7; 95% CI: 0.6–0.9]. Increased age, married women, living in urban residence, and having at least four children were the main risk factors associated with overweight and/or obesity. Conversely, higher education, working in agriculture, and working in manual labor were negatively associated with overweight and/or obesity. Cambodia's non-communicable disease (NCD) public health programs should consider these characteristic for targeting interventions to further reduce overweight and/or obesity in the coming years.

available from the website: https://www.
dhsprogram.com/data/dataset_admin.

**Funding:** The authors received no funding support
for this study.

**Competing interests:** The authors have declared
that no competing interests exist.

**Abbreviations:** BMI, Body mass index; NCDs, Non-
communicable diseases; WHO, World Health
Organization; CDHS, Cambodia Demographic
Health Survey; WRA, Women reproductive age;
AOR, Adjusted odds ratio; PPS, Probability
proportional to size; EA, Enumeration areas.

## Introduction

Obesity or being overweight is associated with increased risk of cardiovascular disease, type 2
diabetes mellitus, systemic hypertension, obstructive sleep apnea, stroke, and some type of
cancers, and maternal disorders among women of reproductive age (WRA) [1,2]. According
to the WHO, overweight and obesity are leading risks for global deaths with at least 2.8 million
adults dying annually due to these conditions [3]. Globally, 2 billion adults aged 18 years and
older were overweight, and 650 million were obese in 2016 [3]. In both developed and develop-
ing countries, obesity or being overweight were higher in women than men; for instance, obe-
sity was 15% in women compared to 11% in men [2,3]. The Global Nutrition Report 2019
showed that 26.1% of women and 20.4% of men were overweight, while obesity was 6.3% in
women compared to 3.5% in men [4]. The final report for the 2104 Cambodia Demographic
and Health Survey (CDHS) indicated that overweight and obesity increased from 6% in 2000
to 18% in 2014 [5]. It was estimated that overweight and obesity contribute to rising healthcare
costs in Cambodia (approximately 1.7% of its annual gross domestic product (GDP) per cap-
ita) and is a major contributor to mortality and decreased general health and productivity [6].

Factors commonly associated with overweight and obesity among women include higher
socioeconomic status, older age, marriage, living in an urban residence, and lack of education
[7,8]. Women with formal employment had higher odds of being overweight or obese than
informally employed women [9]. Also, women who used hormonal contraceptives such as oral
contraceptive pills, implants, patches, and rings were at higher risk of being overweight or
obese [10]. Globally, 30% of daily smokers are overweight or obese [11], with women smokers
at greater risk for overweight or obesity than men smokers [12,13]. While health scholars have
examined these factors internationally, to our knowledge, factors associated with overweight
and obesity specifically among WRA in Cambodia have not been explored. Therefore, we
examined socio-demographic and behavioral factors associated with overweight and/or obe-
sity among WRA.

## Materials and methods

### Ethics statement

The data used in this study were extracted from CDHS 2014 data [14], which are publicly avail-
able with all personal identifiers of study participants removed. Permission to analyze the data
was granted through registering with the DHS program website and submitting an application
outlining the intended use of the datasets [14]. Informed consent was obtained from the study
participants before data collection. The data collection tools and procedures for CHDS 2014
was approved by the Cambodia National Ethics Committee for Health Research (Ref: # 056
NECHR) and the Institutional Review Board (IRB) of ICF in Rockville, Maryland, USA.

### Data sources and procedures

We used the women's dataset from CDHS 2014, which was a nationally representative popula-
tion-based household survey. Survey data were obtained through face-to-face survey inter-
views using a formal survey instrument by trained interviewers. Two-stage stratified cluster
sampling was used to collect the samples. In the first stage, enumeration areas (EAs) stratified
by urban-rural were selected from the sampling frame using probability proportional to cluster
size. In the second stage, a defined number of households, generally 25–30, were selected from
the list of households in each EA using the systematic sampling [5]. Eligible WRA in selected
households were invited to complete a survey interview. In total, 17,578 women aged 15–49
years were interviewed. Weight and height measurements for women were taken by trained

female field staff using standardized instruments and procedures [5]. Height in centimeters (cm), measured using standardized measuring boards with accuracy to 0.1 cm, and weight in kilograms (kg), measured using solar-powered scales (UNICEF electronic scale or Uniscale) with accuracy to 0.1 kg [5]. Data quality assurance and cleaning followed standard operating procedures, which cover every step of the research process, from data collection to data entry [15]. The data were restricted to only include non-pregnant women aged 15–49 years with available body mass index (BMI) data, resulting in the exclusion of 974 pregnant women and 5,786 women with missing BMI data. Data restriction resulted in a final analytic sample of 10,818 WRA.

## Outcome variable

The outcome variable of this study was **Overweight/Obese** for WRA. For the outcome variable **Overweight/Obese**, BMI was first calculated as weight in kilograms divided by the square of height in meters ($kg/m^2$) [16]. BMI was then measured as follows: underweight ($< 18.0 \, kg/m^2$), normal weight ($18.5–24.9 \, kg/m^2$), overweight ($25.0–29.9 \, kg/m^2$) and obesity ($\geq 30.0 \, kg/m^2$) [5]. **Overweight/Obese** was defined as a binary outcome for which women with a BMI $\geq 25.0 \, kg/m^2$ were classified as overweight and obese (coded = 1), while women with a BMI $< 25.0 \, kg/m^2$ were coded as other (coded = 0).

## Explanatory variables

Explanatory variables included socio-demographic characteristics and behavioral factors: Women's age in years (15–19, 20–29, 30–39, and 40–49), marital status (not married, married or living together, and divorced or widowed or separated), education level (no education, primary, secondary and higher), occupation (not working, agriculture, manual labor or unskilled, professional or sealer or services). Households' wealth status was represented by a wealth index calculated via principal component analysis (PCA) and using variables for household assets and dwelling characteristics. Weighted scores divided into five wealth quintiles (poorest, poorer, medium, richer, and richest) each comprising 20% of the population [5,15,16]. Then, the original variable was then recombined into three categories including richer/richest, middle, and poorer/poorest. Residence (urban vs. rural), number of children ever born (no children, one child, two-three children, four and more children), smoking (non-smoker vs smoker), contraceptive usage (not used, hormonal methods (using a pill, Norplant, injection, and vaginal rings), non-hormonal methods (condoms, the diaphragm, the IUD, spermicides, lactational amenorrhea, and sterilization), and traditional methods (periodic abstinence and withdrawal).

## Statistical analysis

Data analyses were performed using STATA version 16 (Stata Corp 2017, College Station, TX) and accounted for the CDHS sampling weights and complex survey design. Socio-demographic characteristics and behavioral factors were described in weighted frequency and percentage. Descriptive analyses that coupled cross-tabular frequency distributions with chi-square tests were used to assess associations between explanatory variables and the outcome variable **Overweight/Obese**. Variables associated with the outcome variable with a significance level of p-value $\leq 0.10$ or that had theoretical justification (for example, women's age, wealth index, and residence) were included in the unadjusted and adjusted logistic regression analyses [17–19].

Unadjusted logistic regression was used to determine the magnitude effect of associations between overweight and obesity with socio-demographic characteristics and behavioral factors

reported as odds ratios (OR) with 95% confidence intervals (CI). Then, adjusted logistics regression was used to assess independent associations, reported as adjusted odds ratios (AOR), with overweight and obesity after adjusting for other independent variables included in the model. Multicollinearity was checked for some original variables including women's age and the number of children ever born, education, and wealth index.

## Results

### Characteristics of the study samples

The mean age of women was 30 years old (SD = 9.9 years) in which age group 20–39 years old accounted for 60.4%. Almost 66.0% were married; 48.4% had completed primary education and 13.6% did not receive formal schooling. A total of 34.2% of women did agricultural work, 21.4% were unemployed. Of the total sample, 36.4% women were from the poor households. The majority (80.7%) of women resided in rural areas. More than one third had two to three children while 31% had no children. Only 2.2% of women reported cigarette smoking, 7.2% reported using hormonal contraceptives. Mean women's BMI was 22 kg/m$^2$ (SD = 3.6 kg/m$^2$) and 15.2% and 2.8% were overweight and obese respectively (**Table 1**).

### Factors associated with overweight and obesity in bivariate analysis

Socio-demographic characteristics were significantly associated with overweight and obesity (**Table 2**). Women had higher prevalence of overweight/obesity if they were aged 40–49 years (31%) compared to younger age groups, (p value <0.001), married (22.9%) compared to other relationship status (p value <0.001), had no education (22.4%) compared to higher education (p value <0.001), were employed in a professional job (28.4%) compared to other jobs, or were from the rich households (21.4%) compared to poor households (p value <0.001). In addition, women who lived in urban areas had a higher prevalence of overweight and obesity than those living in rural areas (22.4% vs 16.9%, p value <0.001). Overweight/obesity increased with parity. Women reported at least four children had a significantly highest overweight and obesity (29.3%), (p value < 0.001). Likewise, women used non-hormonal contraceptives methods had a significant higher proportion of overweight and obesity (27.7%) than women using other methods (p value <0.019).

### Factors associated with overweight and obesity in adjusted logistic regression

As shown in **Table 3**, several factors were independently associated with increased odds of having overweight and obesity among women. These factors included age group 20–29 years [AOR = 2.4; 95% CI: 1.6–3.6], 30–39 years [AOR = 4.6; 95% CI: 3.0–6.9] and 40–49 years [AOR = 6.6; 95% CI: 4.3–10.1] married [AOR = 1.8; 95% CI: 1.3–2.7], women who had professional job [AOR = 1.3; 95% CI: 1.0–1.5], living in urban residence [AOR = 1.3; 95% CI: 1.1–1.5], middle wealth quintile [AOR = 1.4; 95% CI: 1.1–1.8], and rich wealth quintile [AOR = 1.8; 95% CI: 1.5–2.1], having at least two-three children or more. On the contrary, following factors were independently associated with decreased odds of having overweight and obese: women with higher education [AOR = 0.7; 95% CI: 0.6–0.8], working in agricultural jobs [AOR = 0.7; 95% CI: 0.5–0.8], and in manual labor jobs [AOR = 0.7; 95% CI: 0.6–0.9].

## Discussion

Factors associated with being overweight or obese for non-pregnant women include older age, increased parity, marriage, urban residence, higher wealth, and working in a professional

**Table 1. Socio-demographic and behaviors characteristics of the weighted samples of women aged 15–49 years old in Cambodia, 2014 (n = 10,765, weighted count).**

| | Variables | Freq. | % |
|---|---|---:|---:|
| **Women's mean age in years (SD)** | | | 30.0 (9.9) |
| | 15–19 | 1741 | 16.2 |
| | 20–29 | 3553 | **33.0** |
| | 30–39 | 2946 | 27.4 |
| | 40–49 | 2526 | 23.5 |
| **Marital status** | | | |
| | Not married | 2846 | 26.4 |
| | Married or living together | 7096 | **65.9** |
| | Widowed/divorced/separated | 823 | 7.6 |
| **Education** | | | |
| | No education | 1459 | **13.6** |
| | Primary | 5214 | 48.4 |
| | Secondary and higher | 4092 | 38.0 |
| **Current work status** | | | |
| | Not working | 2307 | 21.4 |
| | Agricultural | 3681 | **34.2** |
| | Professional | 2585 | 24.0 |
| | Manual labor and unskilled | 2190 | 20.3 |
| **Wealth index** | | | |
| | Poor | 3915 | **36.4** |
| | Middle | 2064 | 19.2 |
| | Rich | 4786 | 44.5 |
| **Place of residence** | | | |
| | Rural | 8687 | **80.7** |
| | Urban | 2078 | 19.3 |
| **Number of children born** | | | |
| | 0 | 3321 | 30.9 |
| | 1 | 1721 | 16.0 |
| | 2–3 | 3553 | 33.0 |
| | 4+ | 2170 | **20.2** |
| **Smoking status** | | | |
| | Non smoker | 10530 | 97.8 |
| | Smoker | 235 | **2.2** |
| **Ever report of contraceptive use** | | | |
| | No method | 6399 | **59.4** |
| | Traditional method | 2223 | 20.6 |
| | Non-hormonal method | 1370 | 12.7 |
| | Hormonal method | 774 | 7.2 |
| **BMI mean in kg/m² (SD)** | | | 22.0 (3.6) |
| | Underweight | 1501 | 13.9 |
| | Normal weight | 7327 | 68.1 |
| | Overweight | 1637 | **15.2** |
| | Obese | 300 | **2.8** |

occupation. Women with higher education as well as those working in agricultural or manual labor were less likely to be overweight or obese.

**Table 2. Factors associated with overweight and obesity among women aged 15–49 years in Chi$^2$ analysis (n = 10,765, weighted count).**

| Characteristics | | Overweight/Obese (n = 1,937) | | Normal/Underweight (n = 8,828) | | p-value |
|---|---|---|---|---|---|---|
| | | Freq. | % | Freq. | % | |
| **Women's age group in years** | | | | | | |
| | 15–19 | 51 | 2.9 | 1690 | 97.0 | |
| | 20–29 | 390 | 11.0 | 3162 | 89.0 | <0.001 |
| | 30–39 | 705 | 24.0 | 2241 | 76.0 | |
| | 40–49 | 790 | **31.0** | 1735 | 69.0 | |
| **Marital status** | | | | | | |
| | Not married | 159 | 5.6 | 2688 | 94.4 | |
| | Married or living together | 1628 | **22.9** | 5468 | 77.1 | <0.001 |
| | Widowed/divorced/separated | 151 | 18.3 | 672 | 81.7 | |
| **Education** | | | | | | |
| | No education | 327 | **22.4** | 1132 | 77.6 | |
| | Primary | 1051 | 20.2 | 4164 | 79.8 | <0.001 |
| | Secondary and higher | 559 | 13.7 | 3532 | 86.3 | |
| **Current work status** | | | | | | |
| | Not working | 381 | 16.5 | 1926 | 83.5 | |
| | Agricultural | 596 | 16.2 | 3085 | 83.8 | |
| | Professional | 675 | **26.1** | 1910 | 73.9 | <0.001 |
| | Manual labor and unskilled | 285 | 13.0 | 1905 | 87.0 | |
| **Wealth index** | | | | | | |
| | Poor | 543 | 13.9 | 3372 | 86.1 | |
| | Middle | 368 | 17.8 | 1696 | 82.2 | <0.001 |
| | Rich | 1026 | **21.4** | 3760 | 78.6 | |
| **Place of residence** | | | | | | |
| | Rural | 1470 | 16.9 | 7217 | 83.1 | |
| | Urban | 466 | **22.4** | 1612 | 77.6 | <0.001 |
| **Number of children ever born** | | | | | | |
| | 0 | 215 | 6.5 | 3106 | 93.5 | |
| | 1 | 238 | 13.8 | 1483 | 86.2 | <0.001 |
| | 2–3 | 847 | 23.8 | 2706 | 76.2 | |
| | 4+ | 636 | **29.3** | 1533 | 70.7 | |
| **Smoking status** | | | | | | |
| | Non smoker | 1907 | 18.1 | 8623 | 81.9 | 0.108 |
| | Smoker | 30 | 12.7 | 205 | 87.3 | |
| **Ever report of contraceptive use** | | | | | | |
| | No method | 964 | 15.1 | 5435 | 84.9 | |
| | Traditional method | 294 | 21.5 | 1076 | 78.6 | |
| | Non-hormonal method | 214 | **27.7** | 559.2 | 72.3 | <0.001 |
| | Hormonal method | 465 | 20.9 | 1758 | 79.1 | |

Older aged women were more likely overweight or obese as compared to younger women aged 15–19 years. This is consistent with other studies that showed obesity was more prevalent in older WRA [20]. The risk of women becoming overweight or obese rises with age, possibly due to unhealthy food consumption and/or a lack of physical activity [21]. Women with higher education are less likely to be overweight or obesity than women with no school due to

**Table 3. Risk factors associated with overweight and obesity in unadjusted and adjusted logistic regression analysis.**

| Characteristics | Total (N = 10,765) | | | Total (N = 10,761) | | |
|---|---|---|---|---|---|---|
| | OR | 95% CI | P value | AOR | 95% CI | P value |
| **Women's age group in years** | | | | | | |
| 15–19 | 1.0 | Ref. | | 1.0 | Ref. | |
| 20–29 | 4.1 | 2.8–5.8 | <0.001 | **2.4** | **1.6–3.6** | **<0.001** |
| 30–39 | 10.4 | 7.3–14.6 | <0.001 | **4.6** | **3.0–6.9** | **<0.001** |
| 40–49 | 15.0 | 10.6–21.2 | <0.001 | **6.6** | **4.3–10.1** | **<0.001** |
| **Marital status** | | | | | | |
| Not married | 1.0 | Ref. | | 1.0 | Ref. | |
| Married or living together | 5.0 | 4.2–6.1 | <0.001 | **1.8** | **1.3–2.7** | **0.002** |
| Widowed/divorced/separated | 3.8 | 2.9–5.0 | <0.001 | 1.1 | 0.7–1.7 | 0.638 |
| **Education level** | | | | | | |
| No education | 1.0 | Ref. | | 1.0 | Ref. | |
| Primary education | 0.9 | 0.7–1.0 | 0.150 | 0.9 | 0.8–1.1 | 0.495 |
| Secondary and higher | 0.6 | 0.5–0.7 | <0.001 | **0.7** | **0.6–0.8** | **0.001** |
| **Current work status** | | | | | | |
| Not working | 1.0 | Ref. | | 1.0 | Ref. | |
| Professional | 1.8 | 1.5–2.2 | <0.001 | **1.3** | **1.0–1.5** | **0.025** |
| Agricultural | 1.0 | 0.8–1.2 | 0.827 | **0.7** | **0.5–0.8** | **0.001** |
| Manual labor and unskilled | 0.8 | 0.6–1.0 | 0.025 | **0.7** | **0.6–0.9** | **0.011** |
| **Wealth index** | | | | | | |
| Poor | 1.0 | Ref | | 1.0 | | |
| Middle | 1.4 | 1.1–1.7 | 0.009 | **1.4** | **1.1–1.8** | **0.003** |
| Rich | 1.7 | 1.5–2.0 | <0.001 | **1.8** | **1.5–2.1** | **<0.001** |
| **Place of residence** | | | | | | |
| Rural | 1.0 | Ref. | | 1.0 | Ref. | |
| Urban | 1.4 | 1.2–1.6 | <0.001 | **1.3** | **1.1–1.5** | **0.006** |
| **Number of children ever born** | | | | | | |
| 0 | 1.0 | Ref. | | 1.0 | Ref. | |
| 1 | 2.3 | 1.9–2.9 | <0.001 | 1.1 | 0.8–1.7 | 0.505 |
| 2–3 | 4.5 | 3.8–5.4 | <0.001 | **1.5** | **1.1–2.2** | **0.026** |
| 4+ | 6.0 | 4.9–7.3 | <0.001 | **1.7** | **1.1–2.5** | **0.008** |
| **Smoking status** | | | | | | |
| Non smoker | 1.0 | Ref | | - | - | |
| Smoker | 0.7 | 0.4–1.1 | 0.111 | - | - | |
| **Ever report of contraceptive use** | | | | | | |
| No method | 1.0 | Ref | | 1.0 | Ref | |
| Traditional method | 1.5 | 1.3–1.9 | <0.001 | 0.8 | 0.6–1.1 | 0.052 |
| Non-hormonal method | 2.2 | 1.7–2.7 | <0.001 | 1.0 | 0.8–1.3 | 0.769 |
| Hormonal method | 1.5 | 1.3–1.7 | <0.001 | 1.0 | 0.8–1.1 | 0.554 |

**OR:** Unadjusted odds ratios, **AOR:** Adjusted odds ratios, **95% CI:** 95% confidence interval.

increased knowledge and awareness; for example, studies in Nigeria and South Korea reported that educated women had a lower risk of being overweight or obese and that this may be linked to education influencing healthy behavior [22–25]. Education is a critical predictor of women's healthy behaviors and health outcomes, including diet, physical activity [26]. Women with better socioeconomic status and who live in urban areas had higher odds of being overweight and

obese, which is consistent with prior research [27,28]. Women living in urban areas with high income may choose to use cars and other fuel-based vehicles; this convenience results in reduced physical activity and may also increase consumption of fast food. Women with higher socioeconomic status tend to use more improved technologies for a more comfortable lifestyle [29,30]. Women have at least two and more children at risk of being overweight and obese, similar other studies [31]. During pregnancy, factors such as stress, depression, and/or anxiety may play a role in hypothalamic-pituitary-adrenal hyperactivity [32,33]. Women with several children may also have gained weight as a result of their reduced physical activity and have less time to focus on health behaviors including weight management.

This study has limitations. It was a cross-sectional study that does not indicate causality. Additionally, we used to exit secondary data that may result in omitted variable bias as other factors that may be relevant to overweight and obesity such as level of physical activity, dietary intake such as alcohol, sweet food, or soft drink consumption, family history, and awareness about overweight and obesity were not included in the analysis due to data limitations. Finally, these data were gathered in 2014 and may no longer reflect the present conditions of non-pregnant women in Cambodia. A subsequent study should further evaluate these relationships and the general prevalence of overweight and obesity among non-pregnant Cambodian women once additional data becomes available (the next round of data gathering of Cambodia DHS data is currently underway, but has experienced delays due to COVID-19). However, this study has several strengths. First, it draws upon a large, nationally representative sample with a high response rate of 97%. Data were collected using validated survey methods including calibrated measurement tools and highly trained data collectors, which contribute to improved data quality [15]. Formally incorporating the complex survey design and sampling weights into the analysis bolsters the rigor of the analysis and enables generalizing our findings to the population of non-pregnant women in Cambodia.

Our results point to socio-demographic characteristics including older age, urban residence, higher education levels, higher income and wealth, and behavioral factors such as type of employment as risk factors associated with overweight and obesity among women of reproductive age in Cambodia. It is crucial to design intervention programs that target these socio-demographic factors and to raise awareness on the importance of consuming healthy food as well as the benefits of regular physical activity, especially among older women and those with professional jobs.

## Acknowledgments

The authors would like to thank DHS-ICF, who approved the data used for this paper.

## Author Contributions

**Conceptualization:** Samnang Um.

**Formal analysis:** Samnang Um, Jonathan A. Muir.

**Methodology:** Samnang Um, Jonathan A. Muir, Heng Sopheab.

**Project administration:** Samnang Um.

**Writing – original draft:** Samnang Um, An Yom, Heng Sopheab.

**Writing – review & editing:** Samnang Um, An Yom, Jonathan A. Muir, Heng Sopheab.

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
