## [Decision Letter · Decision Letter 0]

12 Dec 2022

PGPH-D-22-01749

Overweight and Obesity among Women at Reproductive Age 15-49 Years Old in Cambodia: Data Analysis of Cambodia Demographic and Health Survey 2014

Dear Dr. Um,

Thank you for submitting your manuscript to PLOS Global Public Health. After careful consideration, we feel that it has merit but does not fully meet PLOS Global Public Health’s publication criteria as it currently stands. Therefore, we invite you to submit a revised version of the manuscript that addresses the points raised during the review process.

In short, the following issues need to address during revision process:

1) Clearly describe the sampling procedure  2) Describe data collection method in details (example: anthropometry) 3) Use reference where appropriate   4) Take care of grammatical issues

We look forward to receiving your revised manuscript.

Kind regards,

Lingkan Barua, MBBS, MPH

Academic Editor

Journal Requirements:

2. Please insert an Ethics Statement at the beginning of your Methods section, under a subheading 'Ethics Statement'. It must include:

a) (for human participants/donors) - A statement that formal consent was obtained (must state whether verbal/written) OR the reason consent was not obtained (e.g. anonymity). NOTE: If child participants, the statement must declare that formal consent was obtained from the parent/guardian.

Additional Editor Comments (if provided):

Reviewers' comments:

Reviewer's Responses to Questions

**Comments to the Author**

1. Does this manuscript meet PLOS Global Public Health’s publication criteria? Is the manuscript technically sound, and do the data support the conclusions? The manuscript must describe methodologically and ethically rigorous research with conclusions that are appropriately drawn based on the data presented.

Reviewer #1: Yes

Reviewer #2: Partly

2. Has the statistical analysis been performed appropriately and rigorously?

Reviewer #1: Yes

Reviewer #2: I don't know

3. Have the authors made all data underlying the findings in their manuscript fully available (please refer to the Data Availability Statement at the start of the manuscript PDF file)?

Reviewer #1: Yes

Reviewer #2: Yes

4. Is the manuscript presented in an intelligible fashion and written in standard English?

Reviewer #1: Yes

Reviewer #2: Yes

5. Review Comments to the Author

Reviewer #1: The greatest strength of the study is that it used a nationally representative sample that made the findings of study valid for Cambodian population. Although the findings of the study is similar to the findings of other international studies of the same population, the result can be used as a base line value for comparison in future.

Some minor revision should be considered. Please see the comment below-

1. Please add reference/s to Methods section for the sampling procedure if there is any published methods for CDHS 2014 survey.

2. Outcome and explanatory variables were defined well in the paper but how the data was collected has not been mentioned explicitly. It was mentioned under the ‘Data Source’ paragraph that participants were interviewed. However, how the body weight and height was measures was not mention in the paper. Please provide details or referred to any published method that was used for the anthropometric measurement.

3. ‘Statistical Analysis’ and ‘Result’ sections were explicit.

4. Grammatical issues:

a) First line of Abstract- you may consider rewriting the sentence as follows-

“Over weight and obesity are the known risk factors for…..”

b) Please consider rewriting the third sentence of the 1st paragraph of the Abstract.

c) In the 2nd sentence of 2nd paragraph of the Abstract- not sure what authors wanted mean by the sentence. Please consider rewriting the sentence.

d) 2nd sentence of 1st paragraph of page 4- Author may consider rewriting the sentence as below- “Overweight and obese women of reproductive age (WRA) group (15-49 years) are at increased risk for ……”

Reviewer #2: First of all, thanks to the author for their efforts. Though it is very common topic worldwide, but the study has few Lackings on its standard. Few common mistakes were observed. Comments on track changes has been made. I expect author will work on that.

6. PLOS authors have the option to publish the peer review history of their article (what does this mean?). If published, this will include your full peer review and any attached files.

**Do you want your identity to be public for this peer review?** For information about this choice, including consent withdrawal, please see our Privacy Policy.

Reviewer #1: **Yes: **Muhammed Shahriar Zaman

Reviewer #2: **Yes: **Fardina Rahman Omi

---

## [Decision Letter · Decision Letter 1]

7 Mar 2023

Overweight and Obesity among Women at Reproductive Age in Cambodia: Data Analysis of Cambodia Demographic and Health Survey 2014

PGPH-D-22-01749R1

Dear Mr. Um,

We are pleased to inform you that your manuscript 'Overweight and Obesity among Women at Reproductive Age in Cambodia: Data Analysis of Cambodia Demographic and Health Survey 2014' has been provisionally accepted for publication in PLOS Global Public Health.

Best regards,

Lingkan Barua, MBBS, MPH

Academic Editor

Reviewer Comments (if any, and for reference):

Reviewer's Responses to Questions

**Comments to the Author**

1. If the authors have adequately addressed your comments raised in a previous round of review and you feel that this manuscript is now acceptable for publication, you may indicate that here to bypass the “Comments to the Author” section, enter your conflict of interest statement in the “Confidential to Editor” section, and submit your "Accept" recommendation.

Reviewer #1: All comments have been addressed

Reviewer #2: All comments have been addressed

2. Does this manuscript meet PLOS Global Public Health’s publication criteria? Is the manuscript technically sound, and do the data support the conclusions? The manuscript must describe methodologically and ethically rigorous research with conclusions that are appropriately drawn based on the data presented.

Reviewer #1: Yes

Reviewer #2: Yes

3. Has the statistical analysis been performed appropriately and rigorously?

Reviewer #1: Yes

Reviewer #2: Yes

4. Have the authors made all data underlying the findings in their manuscript fully available (please refer to the Data Availability Statement at the start of the manuscript PDF file)?

Reviewer #1: Yes

Reviewer #2: Yes

5. Is the manuscript presented in an intelligible fashion and written in standard English?

Reviewer #1: Yes

Reviewer #2: Yes

6. Review Comments to the Author

Reviewer #1: Dear Authors,

The revisions have addressed all of my previous comments for the paper. Congratulation for the nice work.

All the best.

Reviewer #2: Authors have addressed all the comments and i think the draft is now in a good manner.

7. PLOS authors have the option to publish the peer review history of their article (what does this mean?). If published, this will include your full peer review and any attached files.

**Do you want your identity to be public for this peer review?** For information about this choice, including consent withdrawal, please see our Privacy Policy.

Reviewer #1: **Yes: **Muhammed Shahriar Zaman

Reviewer #2: **Yes: **Fardina Rahman Omi
